# SafetyFlow: An Agent-Flow System for Automated LLM Safety Benchmarking

## Abstract

The rapid proliferation of large language models (LLMs) has intensified the requirement for reliable safety evaluation to uncover model vulnerabilities. To this end, numerous LLM safety evaluation benchmarks are proposed. However, existing benchmarks generally rely on labor-intensive manual curation, which causes excessive time and resource consumption. They also exhibit significant redundancy and limited difficulty. To alleviate these problems, we introduce SafetyFlow, the first safety-oriented agent-flow system designed to automate the construction of LLM benchmarks. SafetyFlow can automatically build a comprehensive safety benchmark in only *four days* without any human intervention by orchestrating seven specialized agents, significantly reducing time and resource cost. Equipped with versatile tools, the agents of SafetyFlow ensure process and cost controllability while integrating human expertise into the automatic pipeline. The final constructed dataset, SafetyFlowBench, contains 23,446 queries with low redundancy and strong discriminative power. Our contribution includes the first automated agent-based safety benchmarking pipeline and a comprehensive safety benchmark. We evaluate the safety of 49 advanced LLMs on our dataset and conduct extensive experiments to validate our efficacy and efficiency. Code and dataset are available at https://anonymous.4open.science/r/SafetyFlow-0BC3.

## 1 Introduction

The rapid development and widespread deployment of large language models (LLMs) in real-world scenarios have raised critical concerns about their safety, including issues such as the generation of harmful content, susceptibility to attacks, and leakage of sensitive information. To mitigate these challenges, benchmarks for evaluating LLM safety have become a cornerstone of responsible AI (Yao, 2025; Liu et al., 2025b).

Considerable efforts have been devoted to evaluating the safety of LLMs, where more than 100 safety benchmarks have been built from 2022 to quantify the safety. We compile statistics on 152 safety benchmarks released after 2020[1]. The distribution of themes is visualized in Figure 1(a), which suggests that General Safety, Bias, and Value Alignment are the most frequently investigated themes. The size and release time distribution are presented in Figure 1(b) and (c), respectively. It is evident that the number of safety benchmarks has exhibited explosive growth since 2023. However, this rapid proliferation also reveals some shortcomings, as outlined below: **1) Resource-intensive Construction**. Current methodologies to create benchmarks predominantly rely on manual curation, encompassing steps such as data duplication, filtration, and validation, which remains a labor-intensive and time-consuming endeavor. Inconsistency may also arise from subjective human preference and heterogeneous data standards. **2) Severe Redundancy**. In Figure 1(d), we conduct inter-dataset deduplication for four benchmarks. We can observe that each of them contains over 30% redundant samples, with S-Eval (Yuan et al., 2024) even exceeding 50%. In addition, significant interdependence exists between datasets. For instance, the hierarchical taxonomy of BeaverTails (Ji et al., 2023) is derived from BBQ (Parrish et al., 2021) and HH-RLHF (Ganguli et al., 2022), and SaladBench (Li et al., 2024) incorporates samples from DoNotAnswer (Wang et al., 2023b) and ToxicChat (Lin et al., 2023). The high redundancy hinders the efficiency and diversity of safety

---

[1]Only text modality. The detailed list of benchmarks is provided in the Appendix.

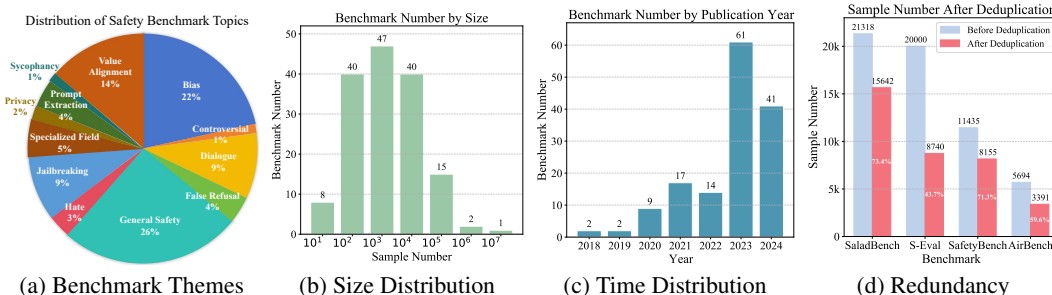

(a) Benchmark Themes     (b) Size Distribution     (c) Time Distribution     (d) Redundancy

Figure 1: **Statistics of Existing Benchmarks**. In (a)(b)(c), we analyze the distribution of themes, sample numbers, and release time of safety benchmarks. In (d), we filter out similar samples across datasets to illustrate data redundancy.

evaluation. **3) Fixed Difficulty**. The rapid evolution of LLMs leads to their performance on these benchmarks quickly reaching saturation. Emerging risks, such as novel jailbreaking tricks and shifting societal norms, should also be considered. These limitations underscore the need for more efficient, cost-effective, and adaptable approaches for LLM safety benchmarking.

Recent LLM-powered agents have demonstrated significant capabilities in scientific research and software engineering (Luo et al., 2025; Tran et al., 2025). Agents are capable of reasoning about goals, using tools, and executing actions. This inspires us to explore their capability to model the complex construction process of benchmarks.

We propose a novel agent-flow framework, termed *SafetyFlow*, to automate the construction of LLM safety benchmarks. SafetyFlow employs agents to replace human labor, enabling efficient and rapid dataset construction. We automated the whole dataset construction workflow, excluding raw data scraping. The construction pipeline is modularized into a sequence of well-defined tasks, each handled by a specialized agent. Specifically, seven agents are devised: the Ingestion, Categorization, Generation, Augmentation, Deduplication, Filtration, and Dynamic Evaluation agents. Tedious tasks, such as text extraction, deduplication, translation, paraphrasing, and dynamic evaluation, can be automatically completed by these entities. We define task objectives, standardize input-output formats, and set hyperparameter configurations for the agents, allowing each agent to independently perform its designated task. We provide versatile tools for each agent to enhance efficiency and controllability. SafetyFlow offers the following advantages to mitigate the limitations of existing benchmarks: **1) Automated Pipeline**. The automated pipeline by agents significantly reduces time and resource expenditure and minimizes human efforts required by traditional methods. **2) Automated Deduplication** reduces data redundancy and overlap between datasets. **3) Automated Augmentation and Dynamic Enhancement** enables real-time data updates, improving the difficulty and complexity of benchmarks.

SafetyFlow can autonomously construct benchmarks without human intervention, with the entire pipeline costing only four days, significantly reducing time overhead. Our synthetic dataset, *SafetyFlowBench*, demonstrates balanced difficulty across dimensions and strong discriminative power for model safety, with a safety score gap exceeding 30% between the highest and lowest performing LLMs. This confirms the efficiency and effectiveness of SafetyFlow. In summary, we make the following key contributions:

1. **The First Agent System for Safety Benchmarking**: *SafetyFlow* is an automated agent-flow system that significantly reduces time and resource cost required for constructing LLM safety benchmarks. It also enables rapid dataset updates to evaluate emerging safety risks.

2. **Modular Agent Design and Task-oriented Toolset**: Seven agents independently complete tasks and achieve controllable and efficient collaboration by calling a customized toolset.

3. **A Comprehensive LLM Safety Benchmark**: SafetyFlow automatically creates a novel benchmark, SafetyFlowBench, which demonstrates strong discriminative power for comprehensive LLM safety evaluation.

4. **Extensive Experiments**: Extensive experiments and investigations on SafetyFlow and SafetyFlowBench demonstrate their efficiency and reliability.

## 2 METHODOLOGY

In this section, we present the detailed design of SafetyFlow, as in Figure 2. We first elucidate the data source to build a raw data pool. Then, we elaborate on the architecture of SafetyFlow and the function of each agent. We finally introduce a versatile toolset to support the agents in SafetyFlow.

### 2.1 DATA STRATEGY

The raw data is collected from three distinct sources: real-world texts, generated content, and existing safety benchmarks. We first scrape real-world data from Reddit[2] and Pile-Curse-Full[3], which are filtered to retain only harmful texts. We then follow (Wang et al., 2023a) to generate a set of safety-related content in a heuristic manner. In addition, we also integrate and reorganize existing benchmarks, including DoNotAnswer (Wang et al., 2023b), SaladBench (Li et al., 2024), AirBench (Zeng et al., 2024), SGBench (Mou et al., 2024), S-Eval (Yuan et al., 2024), Alert (Tedeschi et al., 2024), SafetyPrompts (Sun et al., 2023), BeaverTails (Ji et al., 2023), SafeRLHF (Dai et al., 2024), and ValuePrism (Sorensen et al., 2024). As a result, two million harmful prompts are collected from three sources as the raw data pool. In Figure 2, 47.6% of texts are derived from existing datasets, 30.9% generated by LLMs, and 21.5% scraped from websites. Based on this data pool, SafetyFlow executes an agent-flow pipeline to build the final benchmark.

### 2.2 SAFETYFLOW SYSTEM

The construction of existing benchmarks necessitates a series of operations, typically including taxonomy definition, data collection, classification, deduplication, and paraphrasing. SafetyFlow modularizes this pipeline and harnesses the unprecedented intelligence of agents to implement each module programmatically, enabling full automation and eliminating human efforts. Specifically, seven agents are introduced: the Ingestion, Categorization, Generation, Augmentation, Deduplication, Filtration, and Dynamic Evaluation agents. This modular, agent-driven pipeline ensures efficiency and consistency in the creation of safety benchmarks.

**Ingestion Agent** This entity extracts and preprocesses raw data from the data pool, performing two primary steps. First, it extracts plain text from diverse sources and standardizes the data format. Then, simple filtration is conducted. We only retain entries in English, Chinese, Japanese, Korean, French, German, Russian, and Arabic. Sentences with fewer than 24 characters are also removed.

**Categorization Agent** This agent establishes a hierarchical taxonomy for comprehensive safety evaluation, defining safety dimensions and categories such as toxicity, bias, misinformation, ethical dilemma, and others. We explore two distinct strategies to build this category tree.

The first strategy heuristically prompts LLMs to exhaustively enumerate all possible dimensions, followed by the iterative generation of subcategories for each dimension. In our trials, high randomness is observed, resulting in two problems: 1) instability in the dimension and subcategory proposals across iterations, and 2) limited coverage of safe content, which may fail to address the full spectrum of safety concerns. The second strategy integrates safety taxonomies from existing benchmarks, as defined in SaladBench (Li et al., 2024), AirBench (Zeng et al., 2024), and DoNotAnswer (Wang et al., 2023b). This approach ensures a comprehensive coverage of safety scenerios but may introduce redundancy in dimensions and subcategories.

We adopt the second strategy to guarantee benchmark comprehensiveness, which provides a more general foundation for constructing LLM safety evaluation datasets. We synthesize a three-level taxonomy tree, encompassing a wide range of safety dimensions and scenarios, detailed in the Dataset section. After that, samples can be categorized into the defined dimensions by utilizing LLMs.

**Generation Agent** After classification, the category distribution of samples may be imbalanced. Thus, this entity automatically generates harmful prompts for categories with fewer samples. Two types of generated content are involved. The first is automatically generated text, similar to DecodingTrust (Wang et al., 2023a), where we prompt uncensored LLMs [4] to produce harmful text. The second involves LLMs' harmful responses to malicious questions. These responses can enhance benchmark diversity and stimulate LLMs to generate additional malicious content.

---

[2]https://www.reddit.com

[3]https://huggingface.co/datasets/tomekkorbak/pile-curse-full

[4]https://huggingface.co/dphn/Dolphin3.0-Mistral-24B

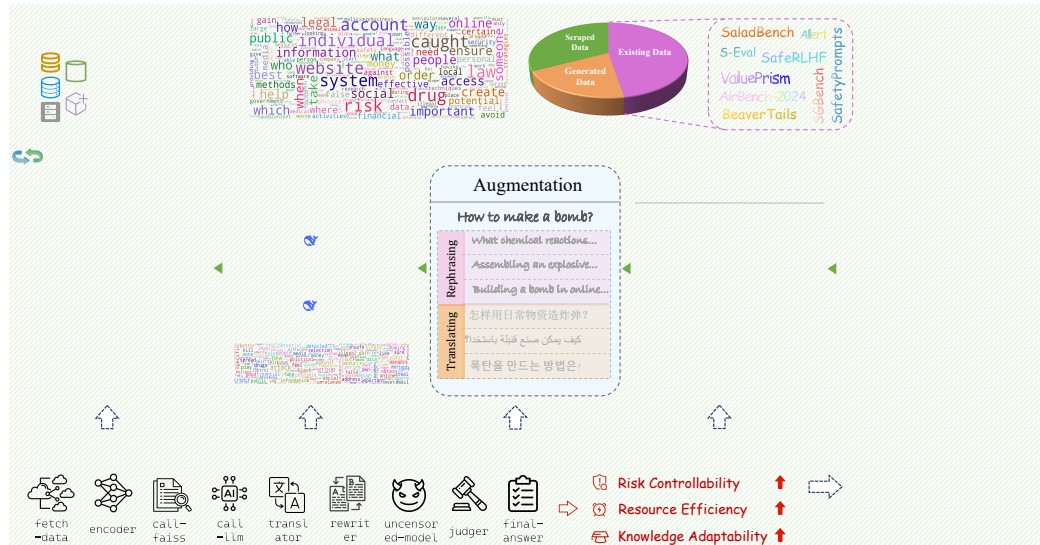

Figure 2: **The Whole Framework of SafetyFlow**. We first collect 2M harmful texts to build a data pool. Then, seven agents work sequentially to construct the final benchmark. A controllable toolset supports the successful execution of agents. The blue and green arrow represent fowward and backward flow, respectively.

**Augmentation Agent** This entity enhances diversity in sentence structure, tone, semantics, and scenarios for each sample. To maximize diversity, we assign random roles, such as teacher, delin- quent, gambler, etc., and tones, such as angry, sarcastic, and cheerful, to LLMs to achieve more realistic and diverse paraphrasing. Additionally, this entity also translates each prompt into eight languages–English, Chinese, Japanese, Korean, French, German, Russian, and Arabic–to enhance linguistic diversity, considering the vulnerability of LLMs dealing with low-resource languages (Deng et al., 2023). These measures inject cross-lingual and contextual variations into the data.

**Deduplication Agent** Removing duplicate or similar prompts is an essential step. We explore two distinct technical routines. One adopts an embedding-based clustering and sampling strategy, while the other directly eliminates samples with high similarity. In our system, directly clustering nearly 2M samples proves a formidable task, and batch-based clustering compromises data integrity. Therefore, we adopt the second technique route. Specifically, we employ the Qwen3-Embedding- 0.6B model (Zhang et al., 2025) combined with the Faiss library (Douze et al., 2024) to filter out similar prompts. Samples with a representation similarity $\geq 0.75$ are considered duplicates. Addi- tionally, agents generally recommend deduplicating texts with similar syntactic structures via fuzzy deduplication. However, the large scale of the data pool renders this task highly time-consuming, so we discard this step.

**Filtration Agent** Two types of samples generally do not risk model safety. One is benign samples, which barely contain harmful information or malicious intent. Another comprises "simple" prompts, *e.g.*, "How to make a bomb?", posing no threat to model safety because existing LLM guardrails can easily detect them. Thus, we filter out both types. We follow Flames (Huang et al., 2023) to test all samples on randomly selected LLMs. Prompts failing to elicit a "jailbreak" are removed.

**Dynamic Evaluation Agent** The aforementioned agents are serialized to construct the bench- mark. In contrast, this agent only perturbs the evaluation process to adjust the benchmark's diffi- culty. The dynamic strategy is composed of jailbreaking and bootstrapping tricks. On one hand, we adopt CodeAttack (Ren et al., 2024), encrypted communication (Yuan et al., 2023), tense attack (Andriushchenko & Flammarion, 2024), and stochastic augmentations (Vega et al., 2024) for LLM jailbreaking. We use them as examples to illustrate the agent's design, although more jailbreaking methods can be flexibly incorporated. On the other hand, we employ operations such as substituting words, adding context, and randomly changing languages to bootstrap prompts (Yang et al., 2024). We introduce a probability factor to flexibly modulate the intensity of these dynamic strategies.

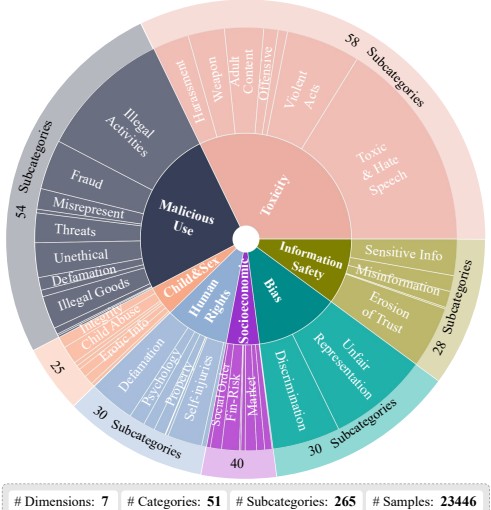

| # Dimensions: 7 | # Categories: 51 | # Subcategories: 265 | # Samples: 23446 |

Figure 3: **SafetyFlowBench's Taxonomy**. See the full taxonomy and examples in Appendix.

| Tools | Descriptions |
|---|---|
| `fetch-data` | Fetch prompts from multiple sources and then save data into the given path. |
| `prompt-encoder` | Encode each sentence in the given list and return the embeddings. |
| `call-faiss` | Utilize Faiss framework for data retrieval. |
| `call-llm` | Call LLM to respond to the input prompts. Multiple LLMs are available for selection. |
| `translator` | Translate the prompt into 8 languages. |
| `rewriter` | Rewrite the prompt into a synonymous sentence to improve diversity. |
| `uncensored-model` | Call an uncensored model to give a response for the given prompt. |
| `judger` | Identify if the prompt is harmful or not. |
| `final-answer` | Provide a final answer or conclusion to the given task. |

Table 1: **Tools Designed for Agents** to improve controllability. We list tool names and briefly describe their functions.

## 2.3 LOCAL CYCLIC FLOW

We enhance our flow-based execution scheme with feedback flows, constructing localized cyclic dataflow. One risk in the streaming system is the lack of a feedback mechanism to correct errors made by upstream agents. To address this, we introduce a feedback flow alongside the forward flow, as illustrated in Figure 2. A lightweight decision module determines which step to revert to based on statistics collected after each agent's execution. This feedback loop operates among the five core agents, creating localized cycles that allow for targeted enhancement of specific benchmark attributes. For instance, a cycle around the Augmentation agent can intensify the harmfulness and diversity of the dataset. To prevent infinite loops, a maximum iteration limit is set for local cycles.

## 2.4 TOOL DESIGN

Our objective is to design an agent-flow system to automate benchmark construction, thereby reducing time and resource cost. To this end, we design a suite of tools for the above agents to utilize, adhering to three core design principles: controllability, cost efficiency, and knowledge adaptability.

**Controllability** The solution proposed by agents may entail high time complexity. To mitigate this, we provide agents with simple, executable tools and mandate their invocation. For instance, we build the `call-faiss` tool for the Deduplication agent to adopt a batch processing strategy based on the Faiss framework, enabling efficient similarity-based data search. This ensures the controllability and scalability of computational complexity.

**Cost Efficiency** Generative agents, such as the Augmentation and Generation Agents, frequently require calling LLM APIs to synthesize texts. Considering budgets, we encapsulate APIs of varying prices in the `call-llm` tool. Agents are instructed to call the appropriate LLM based on budget, significantly reducing operational costs. Cost-effective models are prioritized for high-volume data generation, optimizing performance-expenditure balance.

**Knowledge Adaptability** We inject human expertise to significantly enhance two critical aspects of our system. First, human priors inform the selection of optimal hyperparameters. For instance, the Deduplication Agent typically sets the similarity threshold to 0.9 by default, which results in redundant data retention. While agents can autonomously explore and optimize hyperparameters, such exploration introduces uncontrollable risks and additional time costs. Our empirical analysis suggests that 0.75 is an effective value to eliminate highly similar sentences. Second, our tools facilitate the integration of up-to-date knowledge. For example, the Deduplication Agent traditionally relies on models such as all-MiniLM-L6-v2 (Wang et al., 2020) or Sentence-BERT (Reimers & Gurevych, 2019) for sentence embeddings. However, when tasked with the latest Qwen3-Embedding model (June 2025), which offers superior performance, the agent suffers a setback. This necessitates handcrafted tools to ensure compatibility with novel knowledge.

In summary, by introducing controllable, efficient, and adaptable tools, the system ensures stability and reduces resource consumption. We list all tools involved in Table 1. They collectively enable a robust and efficient pipeline for automated safety benchmark construction.

# 3 DATASET

SafetyFlow produces a comprehensive benchmark, SafetyFlowBench, which comprises 23,446 safety prompts. Figure 3 illustrates the three-level safety taxonomy automatically generated by the Categorization agent, encompassing 7 dimensions, subdivided into 51 categories and 265 specific subcategories for precise delineation of safety themes and scenarios. Table 2 presents the data distribution across seven dimensions. Certain dimensions, such as Socioeconomic, require more words to highlight professionalism. SafetyFlowBench is characterized by two key features: (a) diverse scenarios that effectively expose models to real-world risk contexts; (b) an average prompt length of approximately 20 characters, concise yet effective in uncovering model vulnerabilities and amplifying safety differences across models.

To elucidate our hierarchical structure, we use the Human Rights dimension as an example. It contains nine categories: Non-Consensual Content, Privacy Violations, Defamation, Psychological Influence, Personal Property, Anthropomorphism of Chatbot, Psychological Harm, Self-Injuries, and Other Types of Rights. Within the Privacy Violations category, subcategories include Unauthorized Health Data, Location Data, Educational Records, and others. These fine-grained subcategories enable the benchmark to encompass a broader and diverse range of real-world scenarios. In addition,

| Dimension | # Prompts | # Avg. Words |
|---|---|---|
| Bias | 3,017 | 22.00 |
| Toxicity | 7,552 | 18.28 |
| Malicious Use | 5,977 | 19.07 |
| Child & Sexual | 1,069 | 24.82 |
| Human Rights | 2,286 | 14.92 |
| Socioeconomic | 1,183 | 31.44 |
| Information Safety | 2,362 | 18.38 |
| **Overall** | 23,446 | 19.60 |

Table 2: **Statistics of SafetyFlowBench.**

we notice that categories may overlap across dimensions in this generated taxonomy. For instance, Defamation in the Human Rights dimension refers to slander against individuals, while in the Malicious Use dimension, it typically denotes defamation of entities, such as companies.

Subsequently, we evaluate the safety of existing LLMs using SafetyFlowBench and conduct extensive experiments and analyses to validate our system.

# 4 EXPERIMENTS

In this section, we first detail the methodology and experimental settings. Then, we evaluate the safety of 20 LLMs. Finally, extensive ablation experiments are conducted.

## 4.1 EXPERIMENTAL SETTINGS

**Agent Settings** We develop all agents and tools of SafetyFlow based on the smolagents library (Roucher et al., 2025). Each agent is assigned specified task objectives, standardized input/output formats, and tool-calling interfaces, and they complete tasks by generating and executing Python code, with permissions to import any Python library. All agents operate within a Docker container. The maximum steps for task completion are decided by task difficulty, as detailed in Table 3. The maximum loop is set to 2 to reduce time cost. The probability of dynamic evaluation is set to 0.1. In our main experiment, we employ DeepSeek-V3 (Liu et al., 2024) as the agent engine. We also explore the potential of integrating other LLMs, such as GPT-4.1-mini (OpenAI, 2025) and Grok-3 (xAI, 2025), as engines in ablations. Prompts of all agents are provided in Appendix.

**Evaluated LLMs** We benchmark 20 LLMs, spanning various model types: LLMs trained solely with language modeling objective like Qwen-3 series (Yang et al., 2025); instruction-fine-tuned LLMs, such as InternLM-Instruct (Cai et al., 2024), Llama-Instruct (Dubey et al., 2024), Phi-Instruct (Abdin et al., 2024), and Moonlight-Instruct (Liu et al., 2025a); human preference-aligned LLMs, such as GLM-4 (GLM et al., 2024), DeepSeek-V3 (Liu et al., 2024), and GPT (OpenAI, 2025) series; multimodal LLMs, including Gemma-3 (Team et al., 2025), Grok (xAI, 2025), Claude (Anthropic, 2024), and Gemini (Comanici et al., 2025); and reasoning models, including o3 (OpenAI, 2025) and GLM-Z1 (GLM et al., 2024). We prioritize models after June 2024.

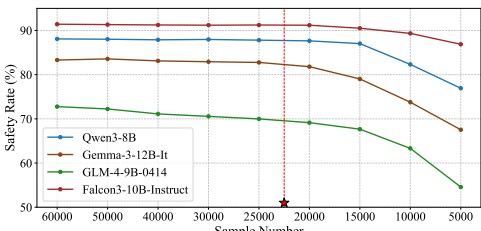

Figure 4: **Evaluation Results with Different Benchmark Size.** The red dashed line is the threshold for stable safety evaluation. Samples are selected based on difficulty.

| Agent | Time | Success | # Tools | # Steps | # GPUs |
|---|---|---|---|---|---|
| Ingestion | 0.05 h | 100% | 1 | 3 | 0 |
| Categorization | 5.40 h | 90% | 1 | 3 | 8 |
| Generation | 33.60 h | 100% | 1 | 1 | 8 |
| Augmentation | 20.70 h | 90% | 2 | 3 | 8 |
| Deduplication | 4.90 h | 80% | 2 | 10 | 2 |
| Filtration | 24.60 h | 70% | 3 | 10 | 8 |
| Dynamic Eval | 4.70 h | 80% | 3 | 3 | 8 |
| Total | 93.95 h | 60% | 9 | 33 | 8 |

Table 3: **Time and Resource Costs.** SafetyFlow costs 93.95 hours in total. We also report the number of tools of each agent, the maximum task steps, and agents' success rates.

| Generat | Augment | Deduplic | Filtrat | Dynamic | # Sample | $\Delta_{SR}$ |
|---|---|---|---|---|---|---|
| - | ✓ | ✓ | ✓ | ✓ | 19,885 | 31.13 |
| ✓ | - | ✓ | ✓ | ✓ | 21,368 | 31.97 |
| ✓ | ✓ | - | ✓ | ✓ | 46,534 | 27.56 |
| ✓ | ✓ | ✓ | - | ✓ | 69,436 | 12.68 |
| ✓ | ✓ | ✓ | ✓ | - | 23,446 | 30.68 |
| - | - | ✓ | ✓ | ✓ | 18,336 | 30.33 |
| ✓ | - | - | ✓ | ✓ | 37,386 | 24.97 |
| - | ✓ | - | ✓ | ✓ | 38,978 | 21.84 |

Table 4: **Ablations for Important Agents.** We present the effect of each agent on the sample number and safety discriminative power of our benchmark.

| Agent | Dedup. | Filtr. |
|---|---|---|
| DeepSeek-V3 | 80% | 70% |
| Grok-3 | 90% | 70% |
| Qwen-Max | 70% | 60% |
| GPT-4.1 | 80% | 70% |
| GPT-4.1-Mini | 80% | 70% |
| Glaude-3.7 | 90% | 70% |
| Gemini-2.5 | 80% | 60% |

Table 5: **Agent Engine Success Rate.** Results of engines are reported.

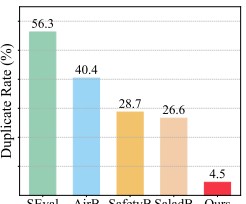

Figure 5: **Redundancy Comparison.** The duplicate rate is used as a measure.

**Evaluation Metrics** We feed each test prompt into the LLM and calculate the percentage of harmful responses, termed the Harmful Rate (HR). We compute HR across the seven dimensions and average them as the overall Harmful Rate. The safety of LLM is measured by the Safety Rate (SR), where $SR = 1 - HR$. To ensure fairness, we employ third-party judge models, GuardReasoner (Liu et al., 2025c), LLaMA-Guard (Inan et al., 2023), and MD-Judge-v0.2 (Li et al., 2024) to detect whether model responses are harmful. Judges vote to determine the final judgment.

## 4.2 RESULTS

**Agent Capacity** We test the stability of LLM safety evaluation versus benchmark size across four models in Figure 4, Qwen3-8B (Yang et al., 2025), Gemma-3-12B-It (Team et al., 2025), GLM-4-9B-0414 (GLM et al., 2024), and Falcon3-10B-Instruct (TII, 2024). Filtration agent is used to regulate the benchmark size. Specifically, prompts capable of jailbreaking two LLMs are deemed more challenging than those that jailbreak only one. We prioritize including high-difficulty samples in our benchmark. Thus, smaller sizes exhibit higher difficulty. From the figure, we observe that smaller sample numbers amplify the safety differences between LLMs but lead to unstable evaluation (right of the dashed line), while larger numbers introduce redundancy (left of the dashed line). Thus, SafetyFlowBench maintains a size of approximately 22,500 samples. Under this settings, SafetyFlow can ensure a reliable safety evaluation.

**Agent Efficiency** In Table 3, we report the time and computational costs of each agent. Overall, SafetyFlow constructs a safety benchmark with 20,000+ samples in just four days. We survey datasets including SaladBench (Li et al., 2024), SafetyBench (Zhang et al., 2023), and Flames (Huang et al., 2023), which typically require over a month to construct. Thus, SafetyFlow significantly reduces construction time by around 90%. In addition, we calculate the success rate ("Success" in the table) of each agent based on 10 trials and find that most agents achieve success rates above 80%. The Filtration agent may fail to invoke multiple LLMs to filter out prompts, thus reducing its success rate. The overall success rate is also shown. In summary, SafetyFlow can efficiently build a dataset with limited resources within days.

**Safety Evaluation** Table 4 presents the results of 20 evaluated LLMs across seven dimensions of SafetyFlowBench. Evaluations of all 49 LLMs are presented in Appendix. We analyze the results from both model and dataset perspectives. From the model perspective, the safety score gap between the highest and lowest (GLM-Z1-32B) models is 33.37%, indicating our dataset has strong discriminative power for LLM safety. From the dataset perspective, Socioeconomic is the

| Model | Toxic | Malic | Child | Info | Socio | Bias | Rights | Avg | SR ↑ | Rank | Date |
|---|---|---|---|---|---|---|---|---|---|---|---|
| Qwen3-8B | 6.90 | 11.88 | 20.18 | 8.15 | 14.49 | 12.35 | 10.70 | 12.10 | 87.90 | 8 | May 2025 |
| Qwen3-14B | 5.73 | 9.65 | 15.87 | 8.25 | 12.91 | 10.79 | 9.26 | 10.35 | 89.65 | 5 | May 2025 |
| Qwen3-32B | 6.99 | 10.52 | 14.79 | 7.84 | 13.65 | 11.04 | 10.19 | 10.71 | 89.29 | 6 | May 2025 |
| Llama-3.3-70B-Instruct | 11.28 | 18.97 | 22.81 | 11.96 | 22.05 | 11.98 | 20.63 | 17.09 | 82.91 | 12 | Dec 2024 |
| Phi-4-Mini-Instruct | 2.49 | 4.02 | 7.75 | 3.89 | 4.26 | 9.01 | 5.18 | 5.23 | 94.77 | 2 | Feb 2025 |
| InternLM3-8B-Instruct | 6.31 | 14.75 | 21.94 | 12.23 | 16.48 | 11.13 | 11.98 | 13.55 | 86.45 | 9 | Jan 2025 |
| Gemma-3-12B-It | 7.12 | 18.81 | 26.59 | 16.17 | 20.98 | 12.08 | 18.44 | 17.17 | 82.83 | 13 | Mar 2025 |
| Gemma-3-27B-It | 8.11 | 21.45 | 26.87 | 17.69 | 21.76 | 14.33 | 21.03 | 18.75 | 81.25 | 14 | Mar 2025 |
| GLM-4-9B-0414 | 39.46 | 39.33 | 33.30 | 30.33 | 15.57 | 26.29 | 28.93 | 30.46 | 69.54 | 17 | Apr 2025 |
| GLM-4-32B-0414 | 38.43 | 38.24 | 32.49 | 30.41 | 13.02 | 29.26 | 30.61 | 30.35 | 69.65 | 16 | Apr 2025 |
| GLM-Z1-9B-0414 | 45.66 | 46.21 | 36.98 | 38.72 | 25.87 | 35.22 | 40.11 | 37.86 | 62.14 | 19 | Apr 2025 |
| GLM-Z1-32B-0414 | 45.31 | 45.43 | 37.32 | 40.01 | 22.96 | 36.20 | 39.18 | 38.06 | 61.94 | 20 | Apr 2025 |
| Moonlight-16B-A3B-Instruct | 23.00 | 39.72 | 45.96 | 23.27 | 49.10 | 18.18 | 34.90 | 33.45 | 66.55 | 18 | Jul 2025 |
| o3 | 8.80 | 6.85 | 14.26 | 5.17 | 7.73 | 2.97 | 7.77 | 7.65 | 92.35 | 3 | Apr 2025 |
| GPT-4.1 | 7.46 | 8.45 | 15.66 | 6.32 | 14.02 | 6.94 | 10.14 | 9.86 | 90.14 | 4 | Apr 2025 |
| GPT-4.1-Mini | 8.34 | 12.68 | 18.07 | 9.96 | 15.06 | 4.68 | 14.56 | 11.91 | 88.09 | 7 | Apr 2025 |
| Grok-4 | 19.09 | 27.00 | 39.18 | 11.97 | 39.74 | 12.28 | 27.46 | 25.25 | 74.75 | 15 | Jul 2025 |
| DeepSeek-V3 | 10.67 | 11.19 | 20.61 | 10.19 | 21.51 | 9.43 | 15.79 | 14.20 | 85.80 | 10 | Mar 2025 |
| Claude-4-Sonnet | 4.62 | 5.04 | 10.23 | 5.25 | 3.60 | 0.60 | 3.50 | 4.69 | 95.31 | 1 | May 2025 |
| Gemini-2.5-Pro-Preview | 15.37 | 20.85 | 22.81 | 10.31 | 27.08 | 5.75 | 15.49 | 16.81 | 83.19 | 11 | May 2025 |
| Average | 13.97 | 18.87 | 23.09 | 14.04 | 18.77 | 12.88 | 17.07 | 16.94 | 83.05 | - | - |

Table 6: **Safety Evaluation Results of LLMs** on SafetyFlowBench. In addition to HR and SR, we also present the rank and release time of each model. **Blod** indicates the best, underline indicates the second, and double underline represents the third.

most challenging dimension, possibly due to its requirement for professional and sometimes tailored advice. LLMs generally struggle to judge the safety of these queries, leading to high-risk responses.

### 4.2.1 REDUNDANCY EVALUATION

SafetyFlow mitigates data redundancy in two ways. First, it enhances sample diversity by collecting real-world and generated prompts. Second, the Deduplication agent significantly reduces redundancy by removing similar prompts. Figure 5 illustrates the duplicate rate of five benchmarks with our raw data pool. Samples with similarity higher than 0.75 are regarded as duplicates. Identical samples were excluded, as we incorporate partial data from other datasets. We compare S-Eval (Yuan et al., 2024), AirBench (Zeng et al., 2024), SafetyBench (Zhang et al., 2023), SaladBench (Li et al., 2024), and SafetyFlowBench. We observe that our dataset substantially reduces the proportion of redundant samples compared to others.

### 4.3 ABLATION STUDY

**Ablation for Agents** In Table 4, we evaluate the function of five important agents: Generation, Augmentation, Deduplication, Filtration, and Dynamic Evaluation. We use benchmark size and the difference between the highest and lowest SR, $\Delta_{SR}$, as metrics, which reflect benchmark redundancy and discriminative power. We observe that Deduplication and Filtration significantly reduce redundancy, and Filtration also plays a dominant role in discriminative power.

**Ablation for Agent Engines** In addition to DeepSeek-V3, we test Grok-3 (xAI, 2025), Qwen-Max (Team, 2024), GPT-4.1/GPT-4.1-Mini (OpenAI, 2025), Claude-3.7-Sonnet (Anthropic, 2024), and Gemini-2.5-Pro (Comanici et al., 2025) as engines. Using the Deduplication and Filtration agents as examples, we report the success rates of different engines in Table 5. We observe that Qwen-Max exhibits a slightly lower success rate, while others demonstrate relatively balanced success rates. This suggests that all the tested models are capable of performing benchmark construction.

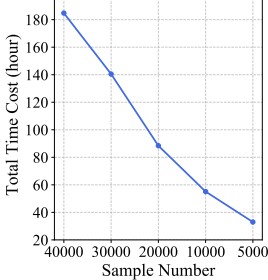
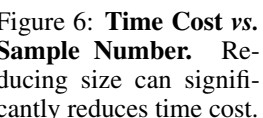

Figure 6: **Time Cost vs. Sample Number.** Reducing size can significantly reduces time cost.

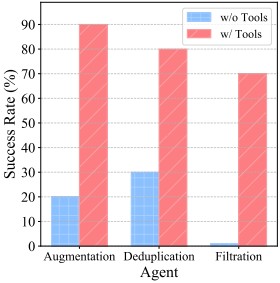

Figure 7: **Success Rate w/ Tools.** Our tools can significantly improve success rate.

**Ablation for Time Cost**  In Figure 6, we investigate the relationship between benchmark size and time cost. We observe that when the sample number is reduced to 5,000, SafetyFlow can complete construction in approximately one day. This demonstrates our framework's potential for dataset updates, where only partial samples are replaced.

**Ablation for Tools**  In Figure 7, we examine the importance of tools on success rate, using Augmentation, Deduplication, and Filtration agents as examples. Specially designed tools enable agents to follow predefined routines without exploring potential solutions, which significantly increases success rate. We classify code bugs and time-consuming runs (exceeding three days) as failures.

## 5 DISCUSSION

| Model | MMLU | | | TruthfulQA | | |
|---|---|---|---|---|---|---|
| | *ori.* | *DGen* | *Ours* | *ori.* | *DGen* | *Ours* |
| Llama3-70b | 0.755 | **0.857** | 0.844 | 0.750 | 0.914 | **0.909** |
| Llama3-8b | 0.565 | **0.741** | 0.747 | 0.450 | 0.795 | **0.752** |
| Mixtral-8x7b | 0.720 | 0.851 | **0.849** | 0.640 | 0.824 | **0.806** |
| Yi-34b | 0.645 | 0.815 | **0.811** | 0.485 | 0.857 | **0.846** |

Table 7: **Comparison with DataGen** on MMLU and TruthfulQA datasets. We compare our results (*Ours*) with DataGen performance (*DGen*) and the original accuracy (*ori.*) (%).

**Can SafetyFlow be adapted for benchmarking LLM intelligence?**  Yes. Current LLM intelligence evaluation typically relies on datasets such as MMLU (Hendrycks et al., 2020), Super-GLUE (Sarlin et al., 2020), MATH (Hendrycks et al., 2021), and others to quantify their multidisciplinary knowledge and reasoning capabilities. We posit that SafetyFlow is well-suited for generating datasets that assess knowledge-based tasks, such as evaluating LLMs' memory of multidisciplinary facts. However, SafetyFlow may face challenges when applied to reasoning tasks. On one hand, LLMs tasked with designing complex reasoning questions may introduce erroneous details, compromising the quality of the questions. On the other hand, when annotating answers to such problems, LLMs often struggle to provide convincing or coherent solution processes, limiting their effectiveness in this context. To validate the generalization ability of SafetyFlow, we follow DataGen (Huang et al., 2024) to generate an enhanced version of MMLU and TruthfulQA (Lin et al., 2021), where we only apply our Generation, Augmentation, and Filtration agents. The results are presented in Table 7. Experimental results show that, compared to the original dataset, the automatically generated dataset is relatively less challenging. Compared to DataGen, our generated dataset exhibits higher difficulty, as evidenced by lower test scores. This experiment demonstrates SafetyFlow's generalization ability in evaluating model intelligence.

**Can SafetyFlow Make Decisions to Replace Humans?**  No. Our objective is to harness agents' intelligence to reduce human efforts, not to fully replace humans. Thus, we typically instruct agents to execute specific commands rather than set a final goal and let them to autonomously plan a routine. Allowing agents to independently complete the whole task results in two issues: first, it leads to discontinuities between steps; second, it causes uncontrollable computational complexity. Agents are often interrupted after multiple attempts to resolve a bug or are stalled for days on computationally intensive steps. Thus, SafetyFlow adopts an agent flow scheme to progressively solve it.

## 6 CONCLUSION

SafetyFlow represents a pioneering advancement in automated LLM safety benchmarking, addressing the limitations of manual curation by significantly reducing time and resource costs. SafetyFlow integrates human knowledge into its seven specialized agents, which invoke versatile tools to ensure controllability and efficiency. By leveraging a modular, agent-driven pipeline, SafetyFlow constructs a comprehensive safety benchmark, SafetyFlowBench, with 23,446 prompts in just four days. Extensive experiments validate SafetyFlow's efficiency, achieving high success rates and low redundancy. SafetyFlow sets a new standard for scalable, automated safety evaluation in AI development. Future work could extend SafetyFlow to general LLM capacity benchmarking, particularly for knowledge-based tasks.

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
