# OpenReview forum: "SafetyFlow: An Agent-Flow System for Automated LLM Safety Benchmarking"
_ICLR.cc/2026/Conference — ICLR 2026 Conference Withdrawn Submission_

### Official Review · Reviewer_ugXE · 2025-10-25

**Soundness:** 3
**Presentation:** 3
**Contribution:** 3
**Rating:** 4
**Confidence:** 3

**Summary:**

The rapid growth of LLMs necessitates reliable safety evaluation to identify vulnerabilities. Existing benchmarks often rely on manual curation, causing time and resource consumption issues. SafetyFlow, a safety-oriented agent-flow system, automates benchmark construction, reducing time and cost. It uses seven specialized agents to create a comprehensive safety benchmark, SafetyFlowBench, with 23,446 queries.

**Strengths:**

+ The paper correctly identifies a major bottleneck in LLM safety research: the labor-intensive, slow, and costly nature of manual benchmark creation.
+ The core idea, using a multi-agent AI system to automate benchmark creation, is innovative.

**Weaknesses:**

- The benchmark is generated entirely by AI agents, which means the data could be too synthetic.
- There are quite a few technical details in the benchmark construction that are not explained well.

**Questions:**

1. Section 2.2: It is not clear why the "seven" agents are introduced. The motivation is not well justified.

2. Section 5: The reasoning capabilities should also be investigated.

3. Section 5: There should be some quantitative metrics to measure how good the argumentation is in classic datasets, e.g., TruthfulQA.

4. Can SafetyFlowBench also generate a benchmark for the agent?

---

### Official Review · Reviewer_xmP6 · 2025-10-28

**Soundness:** 4
**Presentation:** 3
**Contribution:** 3
**Rating:** 4
**Confidence:** 4

**Summary:**

This paper proposes SafetyFlow, a system for automatically constructing LLM safety benchmarks through agent-flow system. The agentic system consists of seven specialized components: Ingestion, Categorization, Generation, Augmentation, Deduplication, Filtration, and Dynamic Evaluation. The system processes 2 million raw data samples and produces SafetyFlowBench, a benchmark containing 23,446 safety queries, in approximately four days. The authors claim this represents a 90% time reduction compared to traditional manual curation methods, while achieving a significantly lower duplication rate of 4.5% compared to existing benchmarks which show 26-56% redundancy. The paper evaluates 20 LLMs on SafetyFlowBench and reports a safety score gap of 33.37% between the highest and lowest performing models, suggesting strong discriminative power.

**Strengths:**

1. The paper presents a clear and well-motivated problem formulation. The authors identify three concrete limitations of existing safety benchmarks with compelling evidence: resource-intensive construction requiring excessive manual labor, severe redundancy with S-Eval showing over 50% duplication and other benchmarks showing 30%+ duplication as demonstrated in Figure 1. This problem definition is particularly timely given the rapid proliferation of safety benchmarks in recent years, and the proposed framework offers practical value for integrating and streamlining benchmark construction.

2. The efficiency improvements demonstrated are substantial and practically significant. The system constructs a comprehensive benchmark in four days compared to approximately one month for traditional methods, representing a 90% time reduction.

3. The system architecture is clearly designed and well-presented. The seven-stage pipeline has well-defined roles and standardized input-output formats for each component. The paper also includes comprehensive evaluation across various LLMs, a detailed taxonomy with 7 dimensions, many categories and subcategories.

**Weaknesses:**

1. One major concern is the absence of quality validation for the automatically generated samples. The paper presents no evidence that the 23,446 automatically generated samples are actually meaningful, semantically coherent, or representative of real safety risks. There is no human expert evaluation of the generated data whatsoever. Domain experts have not assessed whether samples correctly belong to their assigned categories or whether the synthetic data captures genuine safety concerns. The authors need to provide the human evaluation to confirm the quality of the dataset with small samples, and report the portion of low-quality queries.

2. Table 7's generalization experiment can potentially mislead the readers. When applying SafetyFlow to MMLU and TruthfulQA, the authors reported the accuracy of the tasks, interpreting this as "higher difficulty." However, we do not know the performance degradation occured due to the quality degradation of requests or the difficulty of requests. The authors should provide additional evidence to support their claim.

3. What is the definition of the success rate in Table 3? Table 3 reports an overall success rate of only 60%. I would like to know how can we obtain and interpret the value. Are the other 40% of datasets, the failure, discarded? If different runs produce different results due to this 40% failure rate, the system cannot be considered reliably automated.

4. Additional concern on this paper is the agent system design. Despite the authors' claims of presenting an "agent-flow system" with "seven specialized agents," this system is fundamentally a fixed pipeline rather than a genuine multi-agent system. Real agent systems exhibit planning and decision-making capabilities where agents examine goals and determine which agents or tools are needed, but SafetyFlow processes all data through the same seven stages in predetermined order. Genuine multi-agent systems perform selective execution based on data characteristics, but SafetyFlow applies all agents to all samples. Furthermore, the current system does not handle the stream dataset, but just fixed data pool. I acknowledged that each agent is specialized to specified task objectives, standardized input/output formats, and tool-calling interfaces. However, rather than an agentic system perform planning workflow for each request, it is just layers with multiple LLMs to generate dataset.

5. The necessity of the Generation agent is questionable. Why category balancing is actually necessary for evaluating safety benchmark?

**Questions:**

1. What algorithm you used in Faiss to remove duplications?

2. Can the current system suggest different workflow for each data points?

3. What is the reflection model selected to provide the feedback in each flow?

4. How did you choose the hyperparameter 0.75?

**Details Of Ethics Concerns:**

The authors will publicly release 23,446 harmful prompts with documented jailbreaking techniques (CodeAttack, encrypted communication, tense attacks) from Section 2.2's Dynamic Evaluation Agent. While intended for safety evaluation, this creates a comprehensive adversarial attack dataset with no mentioned access controls or usage restrictions. The 265 attack subcategories and proven effectiveness (33.37% safety score variation) make this readily weaponizable against production LLM systems. Moreover, the authors acknowledges scraping Reddit and using Pile-Curse-Full, both containing user-generated content with potential PII (names, locations, personal discussions). The paper mentions no ethical statements on de-identification procedures, privacy impact assessment, or GDPR/CCPA compliance measures, yet plans full public release of this data.

---

### Official Review · Reviewer_L6Wd · 2025-10-30

**Soundness:** 3
**Presentation:** 3
**Contribution:** 3
**Rating:** 4
**Confidence:** 3

**Summary:**

This paper addresses key challenges inherent in existing LLM safety evaluation benchmarks: (1) resource-intensiveness, (2) severe redundancy and (3) fixed difficulty. The authors propose a novel framework that utilizes distinct LLM agents to automate the entire benchmark construction pipeline. This process addresses all stages covered by existing methods, from categorization and generation to deduplication, filtration, and evaluation. This automated agent-flow operates without human intervention and completes the construction process in just 4 days, representing a significant reduction in time compared to traditional methods.

**Strengths:**

1. It automates the human-resource intensive dataset generation process by utilizing separate agents. Its reduced time is impactful compared to other generation methods.

2. It reduces data redundancy by 4.5%, which can hinder the fair evaluation of LLMs. Also, it takes just 4 days to construct the entire benchmark without human labor.

3. Their specialized agents for each stage work harmoniously, and their method covers diverse safety samples, including user characteristics (role, tone, etc.) and languages.

**Weaknesses:**

1. The claim of full automation seems to be overstated. The framework still requires significant manual intervention, particularly for the initial data pool construction and the creation of hand-crafted tools for the agents. Furthermore, the categorization agent's strategy of merely aggregating existing benchmarks raises doubts about the novelty and necessity of the LLM's role in this specific step.

2. The claim that the LLM agent reduces redundancy is unsubstantiated. This improvement appears primarily attributable to the superior embedding model (SentenceBert -> Qwen-Embedding) used for similarity measurement, not the agent's methodology itself. The paper fails to disentangle the impact of the embedding model from the agent's actual contribution.

3. The paper lacks sufficient technical detail to assess its novelty. Key sections, such as Agent and Tool Design (Section 2.4), are described too superficially, obscuring the actual challenges addressed. The Local Cyclic Flow mechanism is also vaguely explained and unconvincing.

4. The claims regarding Agent Capacity (Section 4.2) are insufficiently supported. The experiment's purpose is unclear, and the provided figure (sorting by failure rate) appears to show a trivial outcome. This plot does not adequately justify the conclusions about unstable evaluation or redundancy, nor is the threshold selection process explained.

**Questions:**

**Question**

1. Regarding Figure 1-(d), could you clarify the specific methodology used to calculate the high redundancy (e.g., 30% in SaladBench) of existing benchmarks?

2. Given the fully automated agent-flow, what mechanisms guarantee the quality and coherence of the LLM-generated samples, especially in the absence of human validation?

3. Could you please clarify which models were used for the generation step? There appears to be a discrepancy between the footnote (Dolphin3.0-Mistral-24B) and "Model Bag" in figure 2.

4. Could you explain on the Filtration agent's mechanism for regulating benchmark size? Despite its role, the final categories remain imbalanced, and the specific impact of the Generation/Augmentation agents on this distribution is unclear from the provided ablations.

5. Regarding the Augmentation Agent, could you provide more technical details on the specific mechanisms used to systematically increase the benchmark's difficulty?

6. The agent's suggestion for "fuzzy deduplication" is intriguing. Could you provide more details on this concept and its potential for the agent's self-evolution?

7. When the Generative Agent calls an LLM API, is the monetary budget the sole consideration, or do other factors (e.g., model suitability, task type) influence its decision?

8. The appendix identifies "Phi-4" (Avg 3.19) as the best-performing model, yet it is omitted from the main paper's discussion. Could you explain the reason for this?

**Minor**

1. Line 47. The claim "over 30% redundant samples" appears inaccurate. The provided ratio (15642 / 21318) results in 26.6% redundancy, which is "near 30%" but not "over 30%."

2. Line 374, Incorrect table reference. "Table 4" should likely be "Table 6". The citation on this line also appears broken.

3. Table 5, The bolded value (best result) is DGen (0.857), which is higher than the "Ours" (0.844) result.

4. Figure 3, The caption refers to a "Full taxonomy in Appendix," but this seems to be missing from the appendix.

5. Table 6, Typo in the caption "Blod" should be "Bold".

---

### Official Review · Reviewer_zBAZ · 2025-11-01

**Soundness:** 3
**Presentation:** 3
**Contribution:** 2
**Rating:** 2
**Confidence:** 4

**Summary:**

The paper presents a pipeline for automatically generating safety benchmarks. The pipeline consists of multiple components such as prompts categorization, deduplication and filtering with some of the components employ LLMs in different ways. With the presented pipeline, the paper generates a benchmark of around 23k examples and uses it to evaluate multiple LLMs, showing that the benchmark can significantly differentiate their safety. The paper demonstrates the importance of each component in the pipeline and confirms the time and cost efficiency of the generation process.

**Strengths:**

1. The paper discusses the importance of automating benchmark construction, which is such an important topic with today's rapidly evolving LLMs capabilities.

2. The paper presents a reasonable set of experiments to demonstrate the importance of different design choices, and that the benchmark indeed helps differentiate models in terms of their safety.

3. (Although not explained clearly enough), it is interesting to see that the presented pipeline can be reused for generating benchmarks beyond safety (Table 7).

**Weaknesses:**

1. The paper is limited in novelty and technical depth:
1.1 There already several existing works on automatic benchmark construction for LLM that cover different evaluation angles including safety. While the paper has a related work section (only in the supplementary materials), the paper does not really make a clear distinction between the presented approach and the existing ones. For instance, it is not clear what the advantage of the presented pipeline over Auto-Bencher (https://arxiv.org/abs/2505.22037). Other related work is not covered at all https://arxiv.org/abs/2407.08351 & https://arxiv.org/abs/2406.11939. It is important that the paper clearly explains what makes it a valuable contribution given the existing work.

1.2 It is not clear to me what the presented pipeline is rather than being a sequence of typical data preprocessing operations. The paper needs to highlight the deep technical contributions/insights that are incorporated in the pipeline. I am not even sure why each component is called an "Agent". I don't see any form of autonomy within each component. The processing happening within each component is just a sequence of predefined operations.

2. The description of the presented components is more on the hand-wavy side. For example, how are crowdsourced websites (e.g reddit) are turned into prompts? how are multiple taxonomies from previous work used together to categorize the prompts? Very likely they will contain conflicts and duplicates. How does the augmentation agent ensure the translation quality? especially that in safety cultural aspects should be handled carefully.

**Questions:**

In line 371, "The Filtration agent may fail to invoke multiple LLMs to filter out prompts, thus reducing its success rate".
Would you please explain why that is happening?

---

### Note · Authors · 2025-11-17

I have read and agree with the venue's withdrawal policy on behalf of myself and my co-authors.